

# Development and characterization of a multiplex panel of microsatellite markers for the Reunion free-tailed bat *Mormopterus francoismoutoui*

Muriel Dietrich, Gildas Le Minter, Magali Turpin and Pablo Tortosa

UMR PIMIT (Processus Infectieux en Milieu Insulaire Tropical), Université de la Réunion/ INSERM 1187/C-NRS 9192/IRD 249, Sainte-Clotilde, Réunion Island

## ABSTRACT

The ecology and conservation status of many island-restricted bats remain largely unexplored. The free-tailed bat *Mormopterus francoismoutoui* is a small insectivorous tropical bat, endemic to Reunion Island (Indian Ocean). Despite being widely distributed on the island, the fine-scale genetic structure and evolutionary ecology of *M. francoismoutoui* remain under-investigated, and therefore its ecology is poorly known. Here, we used Illumina paired-end sequencing to develop microsatellite markers for *M. francoismoutoui*, based on the genotyping of 31 individuals from distinct locations all over the island. We selected and described 12 polymorphic microsatellite loci with high levels of heterozygosity, which provide novel molecular markers for future genetic population-level studies of *M. francoismoutoui*.

## INTRODUCTION

About a quarter of over 1,300 currently known bat species are endemic to islands (*Conenna et al., 2017*), in which they sometimes represent the only indigenous mammals. These island-restricted bats play important roles in insular ecosystems through seed dispersal, pollination, and pest control (*Aziz et al., 2017*; *Chen et al., 2017*; *Kemp et al., 2019*). They are also significantly more threatened than continental bat species because of a limited resilience to the combined effects of natural disturbances, typical of island ecosystems, and anthropogenic threats such as urbanization or the development of intensive agriculture (*Jones et al., 2009*). Yet, scientific knowledge of the species biology and conservation status of these island-restricted bats is scarce.

The free-tailed bat *Mormopterus francoismoutoui* (*Goodman et al., 2008*; Fig. 1A) is a small insectivorous tropical bat endemic to Reunion Island, a volcanic island located in the Indian Ocean (Mascarene Archipelago) (Fig. 1B), which has emerged de novo about 3 million years ago (*Cadet, 1980*). This small territory (2,512 km$^2$) is shaped by a very steep mountainous landscape and a great diversity of habitats, which have suffered from the clearing of natural forest, agricultural expansion and urbanization during the 350 years of human colonization (*Lagabrielle et al., 2009*). Despite such a fragmented landscape, *M.*

Corresponding author
Muriel Dietrich,
muriel.dietrich@gmail.com

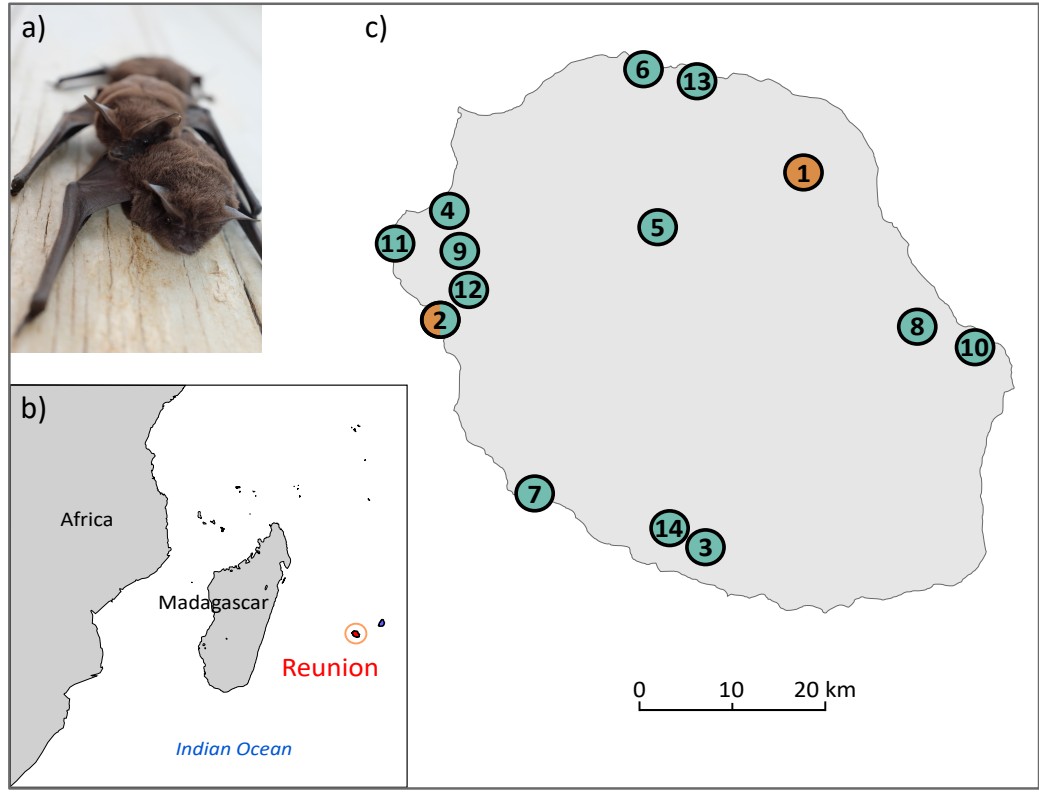

**Figure 1** **(A) Illustration of *M. francoismoutoui* and (B, C) location of bat colonies in Reunion Island used for microsatellite development.** Orange and green circles correspond to colonies where bat organs and wing punches have been collected, respectively. Photo credit: Jean-Marc Le Deun. Maps were created with PanMap and licensed under CC BY 3.0 Unported (*Grobe, Diepenbroek & Siems, 2003*).

*francoismoutoui* is broadly distributed on the island (*Moutou, 1982*; *Barataud & Giosa, 2013*), and forms monospecific colonies that range from a few hundreds to several dozen thousands of individuals (*Dietrich et al., 2015*). This species occupies caves and crevices along with cliff faces but also roosts in a variety of anthropogenic settings, such as buildings, bridges and picnic kiosques (*Goodman et al., 2008*). This proximity with human populations is often considered as a nuisance due to the considerable amount of excrement and their musky smell. As a result, several colonies have been displaced from public spaces (*Augros et al., 2015*). However, *M. francoismoutoui* is not considered at serious threat of population decline Least Concerned (*IUCN, 2019*) but it is protected by French law.

To date, only one study has used molecular markers to evaluate the genetic diversity of *M. francoismoutoui*. *Goodman et al. (2008)* sequenced the mitochondrial D-loop and two nuclear introns (fibrinogen and thyrotrophin encoding genes) from 31 bats sampled in 10 colonies. They found a high genetic diversity using the D-loop marker (28 haplotypes and 2.91% sequence divergence), but only limited genetic structure and no apparent association with geography. However, because of their lower mutation rate compared to nuclear microsatellite markers, mitochondrial markers may be poorly suitable to detect

recent population changes (*Wang, 2010*), such as those induced by landscape changes or the proliferation of anthropogenic roosting sites. Therefore, microsatellite markers represent an ideal tool for inferring such questions (*Cleary, Waits & Hohenlohe, 2016*), especially in the context of *M. francoismoutoui* facing recent habitat fragmentation and agricultural expansion on Reunion Island.

The goal of this study was to use Illumina high-throughput sequencing to develop the first microsatellite markers specifically for *M. francoismoutoui*, and to fully characterize these markers using samples from all over the island. These markers could be used to quantity population genetic structure and evaluate individual movements and dispersal strategies within the mosaic-fragmented landscape of Reunion Island. Future studies can also use these markers to increase understanding of mating behavior, relatedness and paternity, and therefore help designing relevant conservation strategies.

## MATERIALS AND METHODS

### Field sampling

Two types of bat samples were used to develop the microsatellite markers: (1) previously-extracted DNA from organs (pooled kidney, spleen and lung) of 12 bats that were collected in two colonies as part of a previous study (DEAL permit 11/02/2013) (*Mélade et al., 2016*), and (2) wing punches specifically collected for this study from 31 bats in 13 colonies across the island (Fig. 1C, and Supplemental Information 1). For this, bats were captured using harp traps or hand nets. A wing punch of the plagiopatagium was collected using a two mm diameter circular biopsy tool and stored in a sterile vial in an ice cooler in the field, prior to transfer to a −80 °C freezer. Bat capture, manipulation and release protocols were evaluated by the CYROI ethics committee (n °114), approved by the French Ministry of Research (APAFIS#10140-2017030119531267) and conducted under permits delivered by the DEAL (DEAL/SEB/UBIO/2018-09).

### DNA extraction and sequencing

Genomic DNA was extracted from the wing punches using the Qiagen Blood and Tissue Kit. DNA samples (from organs and wing punches) were sequenced by the GENOSCREEN platform, Lille, France (http://www.genoscreen.fr).The DNA quantity was assessed using the Picogreen assay (Invitrogen, Carlsbad, CA, USA). Then, an equimolar pool of 1 to 5 µg of genomic DNA from the 12 bat organs was used for the production of microsatellite libraries, and ran on one lane of an Illumina MiSeq Nano 2 × 300 v2 (Illumina Inc., San Diego, CA, USA).

### Design of primers and genotyping

Data was de-multiplexed and quality-cleaned using Cutadapt v2.1 (*Martin, 2011*) and PRINSEQ (*Schmieder & Edwards, 2011*). Sequences were then merged using Usearch and ran through the QDD V.3.1 software (*Meglécz et al., 2014*). QDD treats all bioinformatics steps from raw sequences until obtaining PCR primers including adapter/vector removal, detection of microsatellites, detection of redundancy/possible mobile element association, selection of sequences with target microsatellites and primer design by using BLAST,

ClustalW and Primer3 softwares (*Thompson, Higgins & Gibson, 1994*; *Johnson et al., 2008*; *Untergasser et al., 2012*). The first step to select primer pairs was done by keeping only perfect di/tri/tetra motives, with A and B quality design (from internal parameters of QDD), and at least 20 bp between each primer and microsatellite sequences, as recommended by QDD.

Among these high-quality loci, 24 (with the longest number of repeats) were then tested for amplification on a limited number of samples (8 wing punches from distinct colonies, Supplemental Information 1). Individual polymerase chain reactions (PCR) were performed in a 10 µL reaction containing 0.5 µL of template DNA (10 ng/µL,) 0.1 µL of FastStartTaq DNA polymerase (Roche, 5U/µL), 1 µL of 10×Buffer, 0.24 µL of DNTPs (10mM), 0.6 µL of $MgCl_2$ (25 mM) and 0.5 µL of each primer (10 µM). Reactions were amplified on a MJ Research PTC-225 Tetrad Thermal Cycler and cycling conditions consisted of 10 min initial denaturation at 95 °C, then 40 cycles at 95 °C for 30s, 55 °C for 30s, 72 °C for 1min, and a final extension step at 72 °C for 10 min. Amplification products (2 µL) were analyzed on a QIAxcel (Qiagen) for the biological validation of primers.

Then, fluorescent labeled forward primers were synthesized (Applied Biosystems) and used with unlabeled reverse primers in a polymorphism analysis performed on a larger set of samples (23 new wing punches from 13 colonies, Supplemental Information 1). We used the same amplification reactions, except that 1 µL of two-fold diluted DNA and between 0.08 µL and 0.25 µL of each primer (20 µM) were used (Table 1). Cycling conditions were the same as for the development step. Amplification products (1 µL) were separated on Applied Biosystems 3730XL Analyzer with GeneScan 500LIZ (Applied Biosystems) internal size standard. The results of the microsatellite profiles were examined using GeneMapper 5 (Applied Biosystems) and peaks were scored by hand. Primer pairs were selected for further analyses when (1) they produced amplicons for all 23 individuals, (2) they did not amplify non-specific fragments, and (3) they revealed length polymorphism (i.e., at least three different allele sizes). For the final selected loci, we genotyped eight additional samples (those tested with the QIAxcel) to reach a total of 31 genotyped samples. These loci were then multiplexed in three reactions using seven samples (Supplemental Information 1) and the same PCR conditions as above.

## Statistical analysis

Linkage disequilibrium for each pair of loci was tested using GENEPOP 4.2 (*Rousset, 2008*) and significance was assessed after a sequential Bonferroni correction. The frequency of null alleles was computed based on the method of (*Brookfield et al., 1996*). We then assessed the number of alleles per locus ($N_A$), the observed ($H_O$) and the expected ($H_E$) heterozygosity per locus, the $F_{IS}$ for each locus as well as the global $F_{IS,}$ using the R package Adegenet (*Jombart, 2008*). We tested whether loci were in Hardy–Weinberg Equilibrium by using 1,000 Monte Carlo permutations, as implemented by the function hw.test in pegas R package (*Paradis, 2010*).

**Table 1  Characteristics of the 12 microsatellite loci developed for *M. francoismoutoui*.** Fluorescent labels attached to forward primers are indicated in brackets.

| Locus name | Repeat motif | Primer sequences (5′–3′) | GB | MP | Range (bp) | NA | $H_0$ | $H_E$ | $F_{IS}$ | $P_{HW}$ | Null$_{all}$ |
|---|---|---|---|---|---|---|---|---|---|---|---|
| MF_loc06 | CA(20) | F: [VIC]ACCCACGACATTCAGCCTTC R: AGAGCTTGGGACCCTGTACT | MN150461 | 1 | 172–188 | 9 | 0.910 | 0.870 | −0.0465 | 0.574 | 0 |
| MF_loc07 | AC(20) | F: [NED]CGCAGCAATTCTCCCAGGA R: CCTTCTGTATAAGGCTGGTGT | MN150462 | 1 | 117–137 | 10 | 0.676 | 0.897 | 0.2427 | 0.042 | 0.067 |
| MF_loc13 | GT(18) | F: [VIC]CTTTCCTCCCTTTCCCGAGG R: GAACCCTCCTTGAGTGAGCC | MN150464 | 1 | 225–250 | 12 | 0.846 | 0.887 | 0.0464 | 0.193 | 0.001 |
| MF_loc15 | TG(17) | F: [6FAM]AGCTCATAATATACCATGCTGACA R: TCTCAGGATGTCTGGCTCCA | MN150466 | 1 | 165–188 | 11 | 0.756 | 0.928 | 0.1847 | 0.170 | 0.065 |
| MF_loc18 | CA(16) | F: [VIC]GCTTAGGGAGCCCTATGTTGT R: GCAAGTGGTTTCTGTTTCTGC | MN150467 | 1 | 122–142 | 10 | 0.897 | 0.848 | −0.0589 | 0.870 | 0 |
| MF_loc03 | GT(21) | F: [6FAM]GGTGGTGTTCTGATACGAGTGT R: TGACAGTTACCCATCCACCC | MN150458 | 2 | 237–255 | 8 | 0.872 | 0.827 | −0.0545 | 0.132 | 0 |
| MF_loc04 | TG(21) | F: [VIC]CCTTGTCTCCTGGCCTCATT R: ACTGTGCCAATTATAATCCTCCC | MN150459 | 2 | 191–229 | 15 | 0.962 | 0.880 | −0.0932 | 0.996 | 0 |
| MF_loc11 | GT(18) | F: [PET]TCTCTGTGGCTGCATCAGTC R: AGAGTCGCATCCAGAAAGATGT | MN150463 | 2 | 280–336 | 13 | 0.897 | 0.846 | −0.0614 | 0.415 | 0 |
| MF_loc28 | AC(12) | F: [NED]GGACTACAGACTTCCGTGCT R: GCTGCCTGGTGAATTGCTTT | MN150468 | 2 | 181–185 | 3 | 0.359 | 0.450 | 0.2011 | 0.578 | 0.041 |
| MF_loc05 | GT(20) | F: [NED]CCAGGAAAGCTGGGTGAAGA R: TGGTTTCCCAGCTCACTTCC | MN150460 | 3 | 260–276 | 8 | 0.756 | 0.747 | −0.0123 | 0.557 | 0.005 |
| MF_loc14 | AC(17) | F: [NED]GACCGAGGCAGGAATAGAGT R: GGCAGGCGAGGCTAAGTTAT | MN150465 | 3 | 298–312 | 8 | 0.897 | 0.844 | −0.0639 | 0.574 | 0 |
| MF_loc36 | TCAT(10) | F: [6FAM]CCAAAGGACTCGCTCGTCTT R: GGCCGTACCCACATTAAATTCA | MN150469 | 3 | 281–309 | 8 | 0.859 | 0.761 | −0.1290 | 0.744 | 0 |

**Notes.**

GB, GenBank accession number; MP, Multiplex locus was assigned to; NA, Number of alleles per locus; $H_0$, Observed heterozygoty; $H_E$, Expected heterozygoty; $P_{HW}$, Hardy-Weinberg *p*-value; Null$_{all}$, null allele frequencies.

## RESULTS AND DISCUSSION

We obtained a total of 1,104,709 sequences from the Illumina run, merged in 58,302 contigs. After the QDD process, 2,378 microsatellite loci were identified, 313 suitable primer pairs were retained and 24 high-quality loci were selected and tested for polymorphism in 23 individuals. Twelve of them showed proper amplification and high level of polymorphism in the 31 genotyped individuals and were multiplexed in 3 distinct reactions. Despite the presence of stutter bands for some loci, none of them impaired the recognition or differentiation between homozygotes and heterozygotes. Primer sequences, the size range of amplification product and multiplex assignment for each of the twelve microsatellite loci are presented in Table 1.

No evidence of linkage disequilibrium was found in the analyzed loci after sequential Bonferroni correction: only 1 pairwise locus combination showed a significant probability of linkage disequilibrium at $p < 0.05$ (MF_loc05-MF_loc11: $p = 0.033$). The percentage of null alleles was low and ranged from 0–6.7%. Global $F_{IS}$ value was 0.0092, suggesting no heterozygous deficiency. Tests for concordance with Hardy–Weinberg equilibrium revealed a marginal deviation in MF_loc07 only (Table 1).

The level of genetic variability was high across loci. Indeed, we found high levels of allelic richness, with an average of 9.6 alleles per locus. Calculation of allele frequencies across the 31 analyzed individuals revealed that MF_loc28 was the only locus with a single allele displaying a frequency greater than 0.6 (Fig. 2). The expected heterozygosity ($H_E$) per locus ranged from 0.450 to 0.928, and the observed heterozygosity ($H_O$) ranged from 0.359 to 0.962 (Table 1). These results are in line with the findings from Goodman et al. (2008) who suggested elevated genetic diversity in the *M. francoismoutoui* population. Based on the analysis of mitochondrial DNA (D-loop), they indeed found a high level of haplotype diversity which was similar to what is found in the closely related species, *Mormopterus jugularis*, inhabiting the much more bigger island of Madagascar (Ratrimomanarivo et al., 2009).

## CONCLUSIONS

In this study, we developed and validated 12 polymorphic microsatellite markers for the Reunion free-tailed bat *M. francoismoutoui*. These markers will facilitate further investigations on population genetics, social structure and behavioral ecology of this bat species. This research can be used to help to develop conservation and management plans for this understudied species. To our knowledge, the markers developed here also represent the first microsatellite markers available for the bat genus *Mormopterus* and may probably be used in closely related species (e.g., *M. acetabulosus* from Mauritius and *M. jugularis* from Madagascar), thus helping clarifying the biogeographic patterns and evolutionary history of these Afro-Malagasy species in islands of the western Indian Ocean.

## ACKNOWLEDGEMENTS

We are grateful to Stéphane Augros and Eco-Med Océan Indien for the help in identifying bat colonies and for the supporting fieldwork provided by Yann Gomard, Steve M.

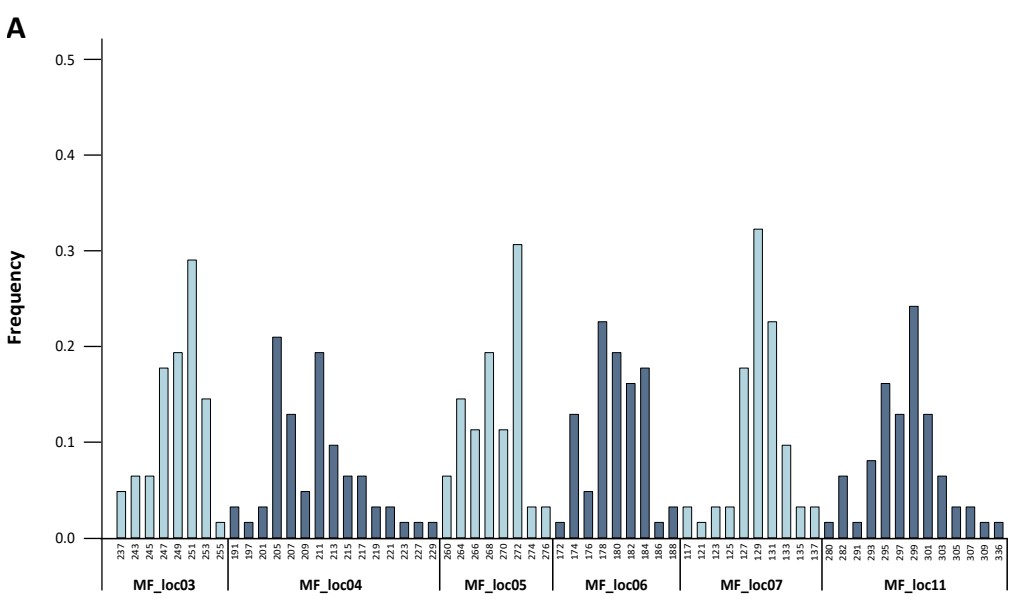

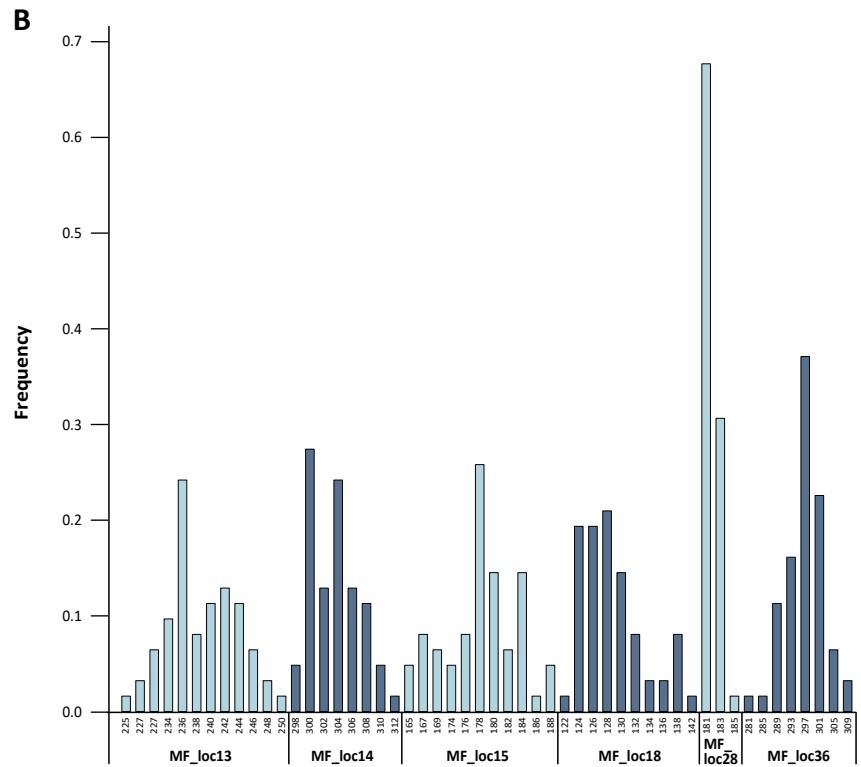

**Figure 2 Baseline allele frequencies for all loci, averaged across all 31 analyzed individuals of *M. fran-coismoutoui*.**

Goodman, Axel Hoarau, Léa Joffrin, Erwan Lagadec, Camille Lebarbenchon, Beza Ramasindrazana, Céline Toty and David A. Wilkinson. We also thank Serena Dool and an anonymous reviewer for their constructive comments on a previous version of this manuscript.

### Funding

This research was supported by the French National Research Agency (ANR JCJC SEXIBAT) and by European Regional Development Funds ERDF PO INTERREG V ECOSPIR number RE6875. The funders had no role in study design, data collection and analysis, decision to publish, or preparation of the manuscript.

### Grant Disclosures

The following grant information was disclosed by the authors:
French National Research Agency (ANR JCJC SEXIBAT).
European Regional Development Funds ERDF PO INTERREG V ECOSPIR: RE6875.

### Competing Interests

The authors declare there are no competing interests.

### Author Contributions

- Muriel Dietrich conceived and designed the experiments, performed the experiments, analyzed the data, contributed reagents/materials/analysis tools, prepared figures and/or tables, authored or reviewed drafts of the paper, approved the final draft.
- Gildas Le Minter and Magali Turpin performed the experiments, approved the final draft.
- Pablo Tortosa performed the experiments, contributed reagents/materials/analysis tools, approved the final draft.

### Animal Ethics

The following information was supplied relating to ethical approvals (i.e., approving body and any reference numbers):

Bat capture, manipulation and release protocols were evaluated by the CYROI ethics committee (no 114) and approved by the French Ministry of Research (APAFIS#10140-2017030119531267).

### Field Study Permissions

The following information was supplied relating to field study approvals (i.e., approving body and any reference numbers):

Bat capture, manipulation and release protocols were conducted under permits 11/02/2013 and DEAL/SEB/UBIO/2018-09, delivered by the Direction de l'Environnement, de l'Aménagement et du Logement (DEAL Réunion).

## Data Availability

Microsatellite sequences and primers of the 12 loci and the genotypes of the 31 bat individuals are available in the Supplementary Files. The sequences are also available at GenBank: MN150458–MN150469.

## Supplemental Information

Supplemental information for this article can be found online at http://dx.doi.org/10.7717/peerj.8036#supplemental-information.

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
