# Peer review of "Development and characterization of a multiplex panel of microsatellite markers for the Reunion free-tailed bat Mormopterus francoismoutoui"

_PeerJ, doi:10.7717/peerj.8036_

## Round 0.1 · original submission · Major Revisions

· Academic Editor

Major Revisions

The previous decision of rejection on the manuscript ‘Development and characterization of a multiplex panel of microsatellite markers for the Reunion free-tailed bat Mormopterus francoismoutoui’ by Dietrich et al. was based on the possibility of plagiarism. As a result, the authors appealed this decision to PeerJ, and I was asked to step in and to provide a final binding decision on the manuscript.

I have read the manuscript, the paper by Cleary et al. (2016), as well as both reviewers’ comments. I concluded that while the authors have paraphrased to a certain extent the sentences of Cleary et al. and that they should clearly have gone further in editing the sentences, they did not reused entirely exact sentences. Importantly, they did not plagiarize concepts, ideas and/or results.

Therefore, I would instead give a major revision decision on this manuscript. I want to re-emphasize the obvious need to modify the Introduction thoroughly to avoid similar problems in the next version. Furthermore, the authors should integrate all other comments provided by both reviewers and, in particular, the comments aiming to help clarify the results of the manuscript regarding the markers developed. Markers information should also be uploaded on GenBank.

· Appeal

Appeal


· · Academic Editor

Reject

One of the reviewers identified possible plagiarism, highlighting similar sentences and paragraphs as in another PeerJ paper by Cleary et al. 2016 (https://peerj.com/articles/2465/). This paper by Cleary et al. 2016 is not cited.

Following PeerJ publication policies: "PeerJ does not tolerate plagiarism, data or figure manipulation, knowingly providing incorrect information, copyright infringement, inaccurate author attributions, attempts to inappropriately manipulate the peer review process, failures to declare conflicts of interest, fraud, and libel. This list is not exhaustive - if there is uncertainty of what constitutes such actions, then more resources may be found at the Committee on Publication Ethics (COPE), the Council of Science Editors (CSE), or the World Association of Medical Editors (WAME)."

I'm therefore obligated to reject your manuscript.

·

Basic reporting

no comment

Experimental design

no comment

Validity of the findings

no comment

Additional comments

In the current study the authors develop novel microsatellites for the tropical bat (Mormopterus francoismoutoui) endemic to the island of La Réunion. M. francoismoutoui was only elevated to species status in 20081 and little work has been conducted on it since. Therefore, the development of this genetic resource is an important and very welcome advance.

The paper is written in a very clear and engaging style and is without any major flaws.
That is, it fulfils the BASIC REPORTING, EXPERIMENTAL DESIGN and VALIDITY OF THE FINDINGS criteria required by PeerJ.

GENERAL COMMENTS

Below are some comments, suggestions and minor typos from the text.

INTRODUCTION

This includes a very good introduction to island bats, to La Reunion and to the species. The goal of the study is clearly stated and some of the many uses of this novel resource are detailed.

Suggestions
Line 57: “However, mitochondrial markers have a much slower mutation rate than nuclear markers,” Rephrase needed: “mitochondrial markers have a much slower mutation rate than nuclear microsatellite markers”. Generally of course the mitochondrial genome has a much higher mutation rate than the nuclear genome. Microsatellites are the exception to this.

Typos/Minor text errors
Line 55: “28 haplotypes and 2.82% sequence..” 2.91% within La Réunion 1

MATERIALS AND METHODS
Typos/Minor text errors
Line 76: “were collected in two colonies in the frame of a”.. Suggested rewrite: “..two colonies as part of a..”
Line 82: “ethic committee”.. ethics committee (add ‘s’)
Line 93: “adapters/vectors removal,” adapter/vector removal, (remove ‘s’)
Lines 102-106: please also detail which Taq you used, how much buffer was added and which PCR machine was used in the amplification.
Line 109: “Then, fluorescent labeled forward primers and unlabeled reverse primers were synthetized”… Probably the reverse primers had already been used at this point in the unlabelled testing of primers? “Then, fluorescent labeled forward primers were synthetized and…”

RESULTS AND DISCUSSION
Suggestions
Line 141-143: LD. What statistical cut off was used to infer there was not LD? Was a correction made for multiple testing (e.g. Bonferroni?) The lines are a little confusing at present as it looks like there is LD between two pairs of loci.
Line 153-154 “These results are in line with the findings from Goodman et al. (2008) (Goodman et al., 2008) who used mitochondrial DNA and found elevated genetic diversity.” Careful here. On the one hand we use the diversity at genetic markers as a proxy for the whole genomic diversity of an organism. Microsatellite loci are a bit of a special case though in that we specifically select the most variable amongst the hundreds of thousands of microsatellite repeat regions in the genome. Thus, when we chose only the most variable, it follows that we should find high variation when using such loci. If we would select microsatellite loci at random across the genome (whether polymorphic or not) then this could be used as an unbiased estimate of genetic diversity and could eventually be used in comparisons.
For the control region, it is hard to say if a certain value is ‘elevated’ or not, as the meaning of this can be vague. Elevated as it is above a certain cut-off value for this most variable part of mtDNA? Elevated compared to other bats, or island bat species for this same region?
Comment:
Line 144-146. It is quite interesting that you tested HWE at the species-level and found almost no deviation. This suggests that not much population structure will be found, and that the species is a single panmictic population.

Typos/Minor text errors
Line 132: “After QDD process”.. “After the QDD process” (add ‘the’)
Line 153-154 The Goodman et al. 2008 reference appears twice.

CONCLUSIONS
Suggestions
Line 159-160: This line returns to the potential uses of this new resource (also outlined nicely at the end of the introduction). All of these things are quite true. However, I am not aware of microsatellite loci availability for this genus at all. If that is so, this could also be mentioned as it is quite a plus.
It is quite likely that these loci will cross-amplify well in closely related taxa (e.g. M. acetabulosus from Mauritius, M. jugularis from Madagascar) as the currently available data suggest these three are very closely related indeed.

These microsatellite loci would be very useful to clarify the biogeography of this genus for the Afro-Malagasy species, their colonisation histories and taxonomy.

Typos/Minor text errors
Line 160: “to help developing conservation” to help to develop conservation

FIGURES & TABLES
Figure 1 is really lovely. Please do add a scale bar for Reunion though.

ANIMAL ETHICS
Typos
Line 187: “ant reference numbers):” ‘and’ reference numbers
Line 188: “ethic” ‘ethics’

SUPPLEMENTAL INFORMATION
-Please also submit the genotypes for the 31 individuals (e.g. as an excel attachment, or genepop file or similar).
-Submitting the sequences of the microsatellites to GenBank will make it much easier for other researchers to find them and eventually reuse them. Also consider submitting the whole illumina run unless there are good reasons not to do so.

Submitting microsatellite sequences in GenBank: https://www.ncbi.nlm.nih.gov/WebSub/
You will need a fasta file containing the sequences and a features file. The feature for your first sequence would be something like this:
>Feature MF_loc03
179 220 repeat_region
rpt_type tandem
rpt_unit_seq "gt"
satellite <microsatellite>

Supplementary table 2: usually the ‘motif’ should match the sequence and not its complement. A mixture of the two seems to be used in this table. Please clarify this.

References
1 Goodman, S. M., Jansen van Vuuren, B., Ratrimomanarivo, F., Probst, J.-M. & Bowie, R. C. K. Specific status of populations in the Mascarene Islands referred to Mormopterus acetabulosus (Chiroptera: Molossidae), with description of a new species. Journal of Mammalogy 89, 1316-1327 (2008).

Reviewer 2 ·

Basic reporting

In general, the article has clear and professional English, structured according to the author guidelines and presents relevant information related to a newly registered endemic bat species in Reunion Island. However, there are some points that should be addressed here, related to a critical lack of references in the introduction and methods section. In the introduction, some lines lack references (lines 32, 35, 40, 45) and should be provided. However, from line 55 up to 75, the structure of the text is pretty similar to Cleary et al. (2016) without mention to this study. Thus, I suggest great modifications in these paragraphs, and proper reference to the authors should be made.
Tables are informative, as well as supplementary materials. The map from sampling areas, on the other hand, is not well descriptive and should be improved, adding elements required in a map. Suggestions are highlighted in the text body.

Experimental design

The study fulfilled the gap in relation to the absence of molecular markers to this species and presented effective and reliable methodology. However, I haven't found GenBank number of sequences and this should be provided according to the journal rules. Few details such as DNA quantification and annealing temperature should also be written in the text. Lastly, sampling coordinates should be provided in supplementary material 1.

Validity of the findings

No comment

Annotated reviews are not available for download in order to protect the identity of reviewers who chose to remain anonymous.

---

## Round 0.2 · Minor Revisions

· Academic Editor

Minor Revisions

I am generally happy with the revisions performed on the previous version. I would suggest a few minor additional edits :

1-Include full loci names (ex. MF_loc07 instead of Locus 07) throughout text : for instance on lines 156, 158, 163 and in figure 2.
2-I could not find the sequence (using the provided accession numbers) on GeneBank – please make sure the GenBank accession numbers in included Table 1 for each loci when resubmitting.
3-Please also include Fis for each loci in Table 1.

---

## Round 0.3 · accepted · Accept

· Academic Editor

Accept

Thank you for the final changes made to the manuscript.